# The Repolarization Period during the Head-Up Tilt Test in Children with Vasovagal Syncope

**DOI:** 10.3390/ijerph17061908

**Published:** 2020-03-15

**Authors:** Ewelina Kolarczyk, Grażyna Markiewicz-Łoskot, Lesław Szydłowski

**Affiliations:** 1Department of Propaedeutics of Nursing, Faculty of Health Sciences in Katowice, Medical University of Silesia, 40-752 Katowice, Poland; 2Department of Nursing and Social Medical Problems, Faculty of Health Sciences in Katowice, Medical University of Silesia, 40-752 Katowice, Poland; mic54@o2.pl; 3Department of Pediatric Cardiology, Faculty of Medical Sciences in Katowice, Medical University in Silesia, 40-752 Katowice, Poland; szydlowskil@interia.pl

**Keywords:** neurocardiogenic syncope, vasovagal syncope, tilt test, electrocardiography, ventricular repolarization

## Abstract

Background: Electrocardiography (ECG) and the head-up tilt test (HUTT) are vital in clinical work-up in children with vasovagal syncope (VVS). Ventricular repolarization parameters (QT) measured during the HUTT can be indicative of electrical instability; however, these parameters are not frequently assessed. This study aimed to investigate if ventricular repolarization parameters measured during the HUTT could be indicative of future ventricular arrhythmias in children with syncope. Methods: The shape and amplitude of the T-wave and parameters of the repolarization period (QT, QTpeak, Tpeak-Tend) were evaluated in a resting ECG performed on the first day of hospitalization and in ECGs performed during three phases of the HUTT. Results: In the after-tilt phase of the HUTT, 19/30 children displayed a change in T-wave morphology. QTc was significantly longer in VVS I compared to that in VVS II patients, but not in the controls (*p* = 0.092). Conclusions: We need further follow-up studies to establish the clinical importance of abnormal dynamics of the repolarization period in children with VVS and negative HUTT. Therefore, children with abnormal T-wave refraction and prolonged duration of the TpTe should remain under the care of a cardiological outpatient.

## 1. Introduction

Syncope accounts for 1–6% of hospital admissions and is not easily diagnosed [1]. Most (~75%) cases of syncope in children and adolescents are caused by an abnormal neuroregulation reflex of the cardiovascular system in an otherwise structurally healthy heart, termed vasovagal syncope (VVS) [1,2,3]. VVS can occur from orthostatic stress (e.g., a long-standing vertical position or sudden change in body position) or emotional distress (e.g., pain, fear), and is characterized by the appearance of prodromal symptoms (e.g., vision and hearing disorders, dizziness, excessive sweating, skin paleness, and dyspnea) [4].

The initial evaluation of syncope is based on careful history taking, followed by a physical examining (including supine and standing blood pressure measurements) and an electrocardiogram (ECG). Once a patient is given a likely diagnosis of VVS, additional specialized tests may be performed, including ECG Holter monitoring, echocardiography, and the exercise stress test. If the possibility of heart disease has been ruled out based on these tests, the head-up tilt table test (HUTT) will be performed [5,6,7]. Although the HUTT test is useful for confirming a diagnosis of VVS, it has limited value in patients with syncope of uncertain cause. Indeed, the latest guidelines from the European Society of Cardiology on the diagnosis and management of syncope recommend that the HUTT should only be used when assessing hypotensive tendency and is not diagnostic of VVS [4].

Despite its limited diagnostic value, changes in repolarization dynamics in ECG recordings during the successive phases of the HUTT, such as the T-wave and the ventricular repolarization period (QT), may be indicative of patient outcomes [8,9]. Indeed, changes in QT parameters due to vertical positioning and orthostatic stress during the successive stages of the HUTT have been suggested to be an important prognostic factor for assessing life-threatening ventricular arrhythmias [10,11]. Unfortunately, these parameters are not routinely assessed in patients.

This study aimed to investigate whether morphological changes in QT parameters recorded on an ECG during the HUTT could be indicative of ventricular arrhythmias in children and adolescents with VVS who showed a negative HUTT result.

## 2. Materials and Methods 

This was a retrospective analysis of the medical records of patients hospitalized between 2013 to 2016 at the Children’s Cardiology Clinic at the Upper Silesian Children’s Health Centre in Katowice, Poland.

### 2.1. The Studied Subjects

Children who met the following criteria were included in the VVS group: a negative family history of sudden cardiac death episodes in relatives of a child before 30 years of age (sudden cardiac deaths in infants, drowning, car accidents in unexplained circumstances), a structurally normal heart, good physical activity tolerance in the exercise test according to the Bruce protocol, no cardiac arrhythmias in the resting ECG and Holter test, normal laboratory results that exclude the presence of inflammation and ionic disorders (i.e., potassium, calcium, magnesium), exclusion of a neurological cause of syncope, and no confirmation of psychogenic pseudosyncope (PPS) in psychological and/or psychiatric examination. For the control group, the inclusion criteria were: children diagnosed with PPS and negative results of the HUTT without cardiological, neurological and endocrine diseases, or the presence of inflammation and ionic disorders. For example, during the test, the children from the PPS group had to report pre-syncope symptoms with no changes in blood pressure, heart rate, or pale skin. The study and control groups were classified by strict selection. In many medical hospitalization records, the study group was isolated, strongly selected, and checked for compatibility of criteria. In total, 30 children diagnosed with VVS and negative HUTT results were included in the study group, and 30 children diagnosed with PPS were included in the control group.

#### 2.1.1. Head-Up Tilt Table Test

The HUTT was carried out according to the Westminster Protocol, without provocative pharmacological tests. In the first phase of the HUTT, the children were made to lay flat in a quiet and warm study room for 30 min. In Phase 2, the children were buckled to the tilt table and lifted to a vertical position (to an angle of 60º). They were kept in that position for 45 min, after which, the child was returned to a flat position. Blood pressure measurements and heart rate evaluation were performed before verticalization, after verticalization, at 5-min intervals throughout the process, and after the child was returned to a flat position.

#### 2.1.2. Electrography

The highest quality ECG records were used for the analysis, and network disturbances were excluded. Records included the standard resting 12-lead ECG recording made on the first day of hospitalization (ECG0), and recordings made during the three phases of the HUTT test, including: the pre-tilt phase (ECG1, after 30 min in a lying flat position); the after-tilt phase (ECG2, after vertical positioning to 60°); and the phase after the HUTT (ECG3, when the child was returned to a flat position, i.e., after 45 min in the upright position).

The determined heart rate (RR intervals), duration of the total repolarization period (the total QT interval), early repolarization period or Q-Tpeak (QTp interval), late repolarization or Tpeak-Tend (TpTe interval), and amplitude and shape of the T-wave were recorded. Measurements were made manually in lead II and V5 in the 12-lead ECG with a paper shift of 50 mm/s and standard amplitude of the feature −1 mV. The analyzed measurements were the average of three consecutive QRS-T evolutions and recorded enlarged with a magnifying glass using a distance stepper. The manual calculation of QTc, Q-Tpeak (QTp interval), and TpTe was done independently by two researchers. The total QT interval was determined from the beginning of the Q-wave to the end of the T-wave, defined as the place where the arm of the descending T-wave returned to the isoelectric line, excluding the U-wave. The QTp interval was measured from the beginning of the Q wave to the peak of the T-wave, whereas the TpTe interval was calculated from the peak of the T-wave to the end of the T-wave. In the case of ‘camel hump’ T-waves, the first peak of the T-wave was used in the measurements [12,13]. The corrected durations of the QT and QTp intervals in relation to heart rate were calculated according to Bazett’s formula.

### 2.2. Statistical Analysis

The results were analyzed statistically using the R environment for statistical analyses, including the PQStat and PSPP programs. The following tests were used to analyze the data: the Chi-squared test, the Student’s *t*-test, the Mann-Whitney U test, the Wilcoxon signed-rank test, the analysis of variance (ANOVA), and the Bonferroni post-hoc test. A *p*-value of <0.05 was considered statistically significant. 

### 2.3. Ethical Approval

Approval from the Bioethical Commission of the Silesian Medical University in Katowice (KNW/0022/KB/116/15) was obtained prior to commencement of this study.

## 3. Results

The duration of the QT, QTp, and TpTe interval and the T-wave morphology was evaluated in the ECGs recorded in children with VVS on admission to the ward and in the three phases of the HUTT. Among the 30 children in the VVS group, 19 (63.3%) showed a change in their T-wave morphology upon verticalization, whereby ‘camel hump’, minus or flat T-waves were recorded in the pre-cordial V4–V6 leads (Figure 1). This group of 19 children showing varying morphology of the T-wave was named the VVS I group. 

In the remaining 11 children (46.7%), the T-wave morphology did not change after verticalization compared to the other phases of the HUTT, except for a slight decrease in T-wave amplitude (termed the VVS II group, no change in the repolarization period) (Figure 2). 

The control group showed normal T-wave morphology in all ECG recordings (Figure 3).

### 3.1. Characteristics of the Study Groups

The average age of the children with VVS was 16.2 ± 1.3 years, and the majority were female (22/30 or 73%). The average age of children in the control group was 14.7 ± 2.3 years, and again, most were female (22/30 or 73%). The VVS I and VVS II groups did not differ significantly in terms of age: 16 ± 1.3 years vs. 16.4 ± 1.3 years; however, children from the control group were statistically significantly younger than those from both VVS groups.

Children experienced their first episode of syncope (fainting) at age 15.5 ± 1.4 years in the VVS I group and at 14.5 ± 2.8 years in the VVS II group. Children from the control group had their first fainting episode at a significantly younger age, 13.8 ± 2.5 years, than those in the VVS I group.

In terms of the prodromal symptoms heralding syncope, a feeling of weakness was more frequent in the VVS II group than in the VVS I group. Skin paleness occurred only in the VVS I group. Headache was significantly more frequent in both VVS groups compared to the control group. The characteristics of the children in the three groups (VVS I, VVS II, and control) are summarized in Table 1.

### 3.2. T-wave and Repolarization Parameters

There were no statistically significant differences between the II and V5 leads in terms of the T-wave amplitude or the duration of the repolarization parameters. Overall, the RR, QT, QTc, QTc, QTp, QTpc, and TpTe intervals recorded during ECG3 were similar to those from ECG1, indicating a return to the initial values after the changes during phase two of the HUTT. 

The lowest T-wave amplitudes in leads II and V5 were observed during the after-tilt phase of the HUTT. In the VVS I group, the T-wave amplitude was lowest in lead V5 during the after-tilt phase (1.9 mm), and significantly different to the VVS II (2.8 mm) and control (3.0 mm) groups. During the pre-tilt phase (ECG1), the T-wave amplitudes in both VVS groups (I and II) were significantly different from that of the control group.

In terms of the repolarization parameters, the shortest RR interval was recorded during the after-tilt phase of the HUTT (ECG2) in all three groups (Table 2) and was significantly different from the RR intervals recorded during the three other phases (i.e., ECG0, ECG1, or ECG3). However, the RR values were not significantly different between the VVS I, VVS II, or control groups in the individual phases of the HUTT. The repolarization parameters are shown in Table 2.

There were no significant differences in the QT interval durations among the three groups during the individual phases of the HUTT (Table 2). The shortest QT interval duration was recorded during the after-tilt phase of the HUTT (ECG2) in all three groups. The QT interval duration was significantly longer during ECG1 compared to ECG0 in all groups.

The longest Bazett-corrected QT intervals (QTc) were recorded during the after-tilt phase of the HUTT (ECG2) in all three groups (Table 2). The QTc measurements recorded during ECG2 were significantly different from the QTc recorded during ECG0 (in all groups), during ECG1 (in the VVS I and control groups), and during ECG3 (only in the control group). The duration of the QTc interval measured during ECG1 was similar to that recorded during ECG3 and was comparable across all three groups. However, the duration of the QTc interval recorded during ECG1 was higher than that recorded during ECG0 in all three groups. The duration of the QTc interval in the VVS I group recorded during the after-tilt phase was significantly longer than that in the VVS II group (451.3 ms vs. 434.4 ms, *p* = 0.004), but was not significantly different to the control group (442.3 ms, *p* = 0.092).

The shortest QTp interval was recorded during ECG2 in all three groups (Table 2). The QTp measurements recorded during ECG2 were significantly different from those recorded during ECG0 (only in the VVS I group), during ECG1 (in all groups), and during ECG3 (only in the VVS I group). The QTp interval was significantly longer during ECG1 than ECG3 in the VVS I group, and significantly longer during ECG1 than ECG0 in the control group. Furthermore, the duration of the QTp interval during ECG2 was significantly longer in the control group (283 ms), compared to the VVS I (247.4 ms) and VVS II (260.9 ms) groups.

The QTp interval corrected by Bazett’s formula (QTpc) was comparable between the VVS I and VVS II groups, and was not significantly different during ECG0, ECG1, ECG2, or ECG3 (Table 2). In the control group, the longest QTpc interval was observed during the after-tilt phase of the HUTT (ECG2). In all three groups, the QTpc interval was longer during ECG1 compared to ECG0. Moreover, the duration of the QTpc interval was significantly shorter during ECG2 in the VVS I (321ms) and VVS II (324.3 ms) groups, compared to the control group (359.4 ms).

The longest duration of the TpTe interval in the VVS I group was recorded during ECG2 (100 ms) compared to ECG0 (90.5 ms), ECG1 (91.1 ms), and ECG3 (92.6 ms) (Table 2). In addition, the TpTe interval in the VVS I group was longer than that observed in the VVS II group (88.2 ms) and the control group (65 ms). The TpTe interval did not vary significantly across the four ECG records in the VVS II group. Meanwhile, the TpTe interval was significantly shorter in the control group during ECG2 compared to ECG0, ECG1, or ECG3, and also compared to that recorded for the VVS I and II groups during all ECG recordings. In all groups of children (VVS I, VVS II, and control), the duration of the TpTe interval recorded during ECG3 was comparable to that recorded in ECG1.

## 4. Discussion

This study found abnormal dynamics of the repolarization period together with abnormal T-wave morphology and prolonged duration of the TpTe in children with VVS and negative HUTT results. Similar observations concerning changes in the T-wave morphology were made by Mayuga and colleagues, in which the T-wave in the V5 lead became flat in 39% of the patients, and even became negative in 21% of the patients, during a negative HUTT. Moreover, these T-wave changes persisted during the after-tilt phase of the HUTT in those with a positive vasovagal reaction [14]. However, Mayuga and colleagues did not observe any ‘camel hump’ T-waves in these patients, which are a characteristic morphological disorder of the repolarization period in those with long QT syndrome (LQTS). A previous study showed that the morphology of T-waves in the V4–V6 leads can be used to identify the LQTS genotype (i.e., T-waves are usually flat or ‘camel hump’ shaped in about 66–88% of carriers of the LQTS2 genotype) [15]. Moreover, Takenaka and colleagues showed that patients with LQTS2 can display T-waves with normal morphology and duration in resting ECG, which later become ‘camel hump’ shaped during an exercise stress test [16]. Similarly, in this study, the T-waves changed morphology to either flat or ‘camel hump’ shaped in 19 children with VVS during the HUTT.

In terms of the QT parameters, Walker et al. showed that after verticalization during an exercise test, the corrected QT interval was significantly longer in 16 patients with LQTS2 compared to a healthy control group [17]. Similarly, Hermosillo and colleagues observed an increase in the corrected QT interval in patients with prolonged QT syndrome (with a positive vasovagal response during the HUTT), two minutes before the onset of fainting, without changes in the morphology of T-wave refraction [18]. In this study, the corrected QT interval recorded during the after-tilt phase in children with VVS with morphological changes of T-wave refraction was significantly longer than those without such T-wave changes (451.3 ms vs. 434.4 ms, respectively); however, it was not significantly longer to the QTc recorded in the control group.

The RR interval was shortened during the after-tilt phase of the HUTT in all children with VVS in this study, although the heart rate was faster in those with morphological changes of T-wave refraction compared to those without (650.9 ms and 595.3 ms, respectively). Findler et al. also found that the RR interval (825 ms vs. 712 ms) and QT interval (363 ms vs. 354 ms) was shortened during the tilt phase in children with syncope compared to at rest, with an increase in the corrected values of the QTc interval (402 ms vs. 423 ms) [19]. Another study showed accelerated cardiac rhythm with QTc prolongation and ventricular arrhythmia in 13 children with VVS and negative HUTT results after the administration of isoproterenol [20]. Meanwhile, in this study, the RR interval was accelerated during the after-tilt phase of the HUTT in all patients without pharmacological intervention. Moreover, the QT values were comparable across all three groups (i.e., VVS I, VVS II, and control), with the changes in QTc values corresponding to the different heart rates (i.e., the shortest RR was observed in the VVS I group).

The TpTe interval was significantly longer in children with VVS and variable T-wave morphology compared to those without T-wave morphology changes (100 ms vs. 88.2 ms, respectively) and the control group during the after-tilt phase of the HUTT. Similarly, Buttà and colleagues reported that the TpTe and QTc intervals in the after-tilt phase were significantly longer in patients with VVS compared to the control group [21]. The TpTe interval, which shows the global dispersion of repolarization, is considered to be a diagnostic arithmetic index, especially when the T-wave changes into a ‘camel hump’ or biphasic shape [22,23].

Previous studies have indicated that variations in ECG repolarization parameters arising from verticalization and orthostatic stress during the successive stages of the HUTT may be an important prognostic factor in the assessment of life-threatening symptomatic ventricular arrhythmia in patients with syncope. This study found that the QTc time was significantly longer in children with abnormal T-wave morphology than in those without T-wave changes, primarily due to the prolongation of the TpTe. Indeed, TpTe prolongation is considered to be responsible for the dispersion of repolarization, and therefore, has a potential pro-arrhythmic effect [10,12,14]. Therefore, despite its low sensitivity (26–80%) and lack of diagnostic utility [4,24], the HUTT may be a useful prognostic tool in assessing the hidden risk of ventricular arrhythmias in children. Indeed, the stratification of the risk of sudden cardiac death and life-threatening events based on the presence of structural and primary heart disease and ECG abnormalities described in our paper is in accordance with the latest guidelines of the European Society of Cardiology 2018 [4].

The limitations of this work include the small sample size of the analyzed group (*n* = 30), which arose from the strict inclusion and exclusion criteria required to avoid all possible factors influencing prolongation of the repolarization period. Other limitations of this work include the methodological limitations of repolarization measurements. To alleviate any potential issues, only the highest quality electrocardiograms were included in the study, without network disturbances, and the repolarization parameters were measured using a magnifying glass on a pedometer. Moreover, the techniques used to determine the tangent to the descending arm of the T-wave and the evaluation of the differences between the longest and shortest QTs interval (QT dispersion) were not performed in this study, as some experts speculate this may result in underestimation of the QT interval [13,25]. Finally, this work focuses on the potential predispositions to arrhythmia, not the occurrence of arrhythmia itself, and therefore, it is difficult to make conclusions about direct links between ECG changes and patient outcomes. This research is only a small step in expanding the knowledge in this area, but it is a first “building block” in this direction.

## 5. Conclusions

We need further follow-up studies to establish the clinical importance of abnormal dynamics of the repolarization period in children with VVS and negative HUTT results. From a clinical standpoint, children with VVS and negative HUTT results, who are diagnosed with after-tilt repolarization abnormalities in ECGs, require systematic and frequent check-ups in cardiology out-patient clinics, in addition to standard prophylactic recommendations with lifestyle modifications.

## Figures and Tables

**Figure 1 ijerph-17-01908-f001:**
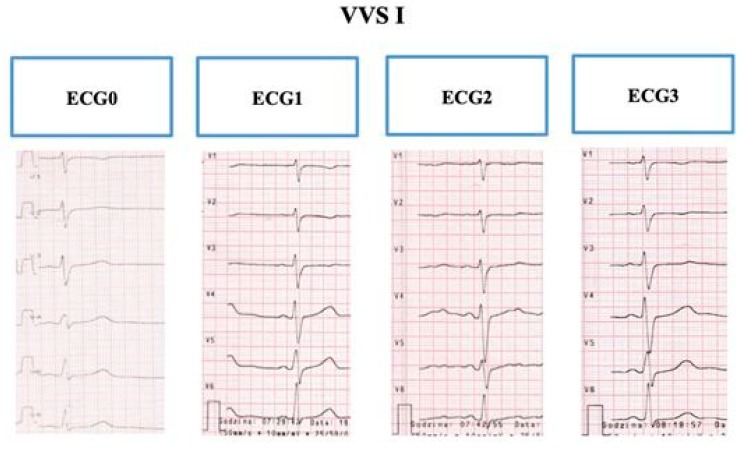
The sample electrocardiogram of a child with vasovagal syncope with variable T-wave morphology (VVSI) recorded on admission to the ward and in the three phases of the tilt test: two-humid T-waves are visible in a 2-phase electrocardiogram (after standing) in V4-V6 leads. Source: the medical documentation of a child with children with vasovagal syncope, hospitalized at the Children’s Cardiology Clinic.

**Figure 2 ijerph-17-01908-f002:**
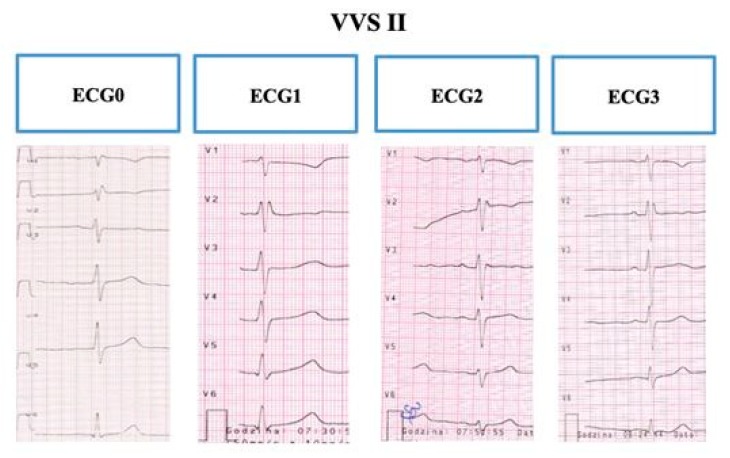
The sample electrocardiogram of a child with vasovagal syncope with T-waves normal morphology recorded on admission to the ward and in the three phases of the tilt test: in all electrocardiogram recordings, T-waves did not change after verticalization. Source: the medical documentation of a child with vasovagal syncope, hospitalized at the Children’s Cardiology Clinic.

**Figure 3 ijerph-17-01908-f003:**
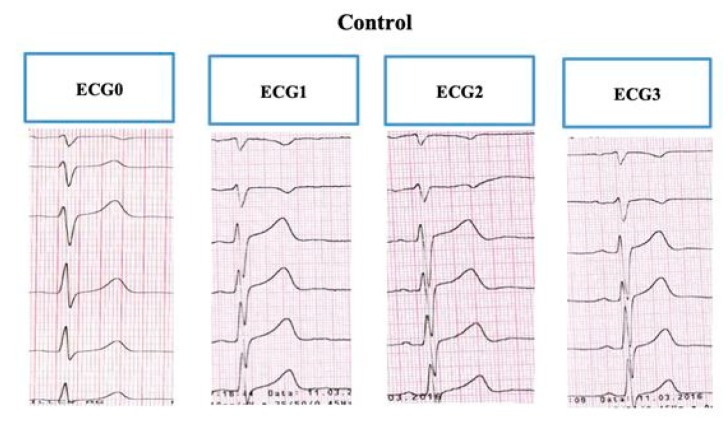
The sample electrocardiogram of a child from the control group recorded on admission at the branch and in the three phases of the tilt test: all electrocardiogram recordings had normal T-wave morphology. Source: the medical documentation of a child with psychogenic pseudosyncope, hospitalized at the Children’s Cardiology Clinic.

**Table 1 ijerph-17-01908-t001:** Characteristics of children with vasovagal syncope, either with or without morphological changes on the electrocardiogram, compared to those with psychogenic pseudosyncope (control).

Characteristics	VVS I ^1^ (*n* = 19)	VVS II ^2^ (*n* = 11)	Control ^3^ (*n* = 30)
Age (at time of hospitalization), years	16 ± 1.3 ^*^	16.4 ± 1.3 ^*^	14.7 ± 2.3
Male, *n* (%)	5 (26.3%)	3 (27.3%)	8 (26.7%)
Female, *n* (%)	14 (73.7%)	8 (72.7%)	22 (73.3%)
Age at which the child had their first episode of syncope	15.5 ± 1.4 ^*^	14.5 ± 2.8	13.8 ± 2.5
**Circumstances of syncope**			
After effort, *n* (%)	8 (42.1%)	7 (63.6%)	13 (43.3%)
Emotions/stress, *n* (%)	8 (42.1%) ^*^	3 (27.3%) ^*^	1 (3.3%)
Change of body position, *n* (%)	13 (68.4%) ^*^	6 (54.5%) ^*^	3 (10%)
Long-term verticalization, *n* (%)	10 (52.6%)	6 (54.5%)	7 (23.3%)
Lying and sitting position, *n* (%)	0 (0%)	0 (0%)	2 (6.7%)
**Prodromal symptoms**			
Headache, *n* (%)	8 (42.1%) ^*^	5 (45.5%) ^*^	4 (13.3%)
Dizziness, *n* (%)	7 (36.8%)	4 (36.4%)	7 (23.3%)
Weakness, *n* (%)	4 (21.1%)	8 (72.7%) ^*^	3 (10%)
Hands tremor, *n* (%)	2 (10.5%)	2 (18.2%)	1 (3.3%)
Pale skin, *n* (%)	3 (15.8%) ^*^	0 (0%)	0 (0%)
Sweating, *n* (%)	2 (10.5%) ^*^	0 (0%)	1 (3.3%)
Tinnitus, *n* (%)	2 (10.5%)	2 (18.2%)	1 (3.3%)
Blurred vision, *n* (%)	5 (26.3%)	5 (45.5%)	6 (20%)
Dyspnea, *n* (%)	5 (26.3%)	3 (27.3%)	7 (23.3%)
Palpitations, *n* (%)	3 (15.8%)	2 (18.2%)	8 (27.7%)
Chest pain, *n* (%)	3 (15.8%)	1 (9.1%)	8 (26.7%)
Numbness in the limbs, *n* (%)	0 (0%)	0 (0%)	3 (10%)
Hot feeling, *n* (%)	0 (0%) ^*^	0 (0%) ^*^	1 (3.3%)
Nausea, *n* (%)	0 (0%) ^*^	0 (0%) ^*^	3 (10%)
Fatigue, *n* (%)	0 (0%) ^*^	0 (0%) ^*^	3 (10%)

^1^ VVS I: children with vasovagal syncope (VVS) and a change in their T-wave morphology during the head-up tilt table test. ^2^ VVS II: children with VVS and no change in their T-wave morphology during the head-up tilt table test. ^3^ Control: children with psychogenic pseudosyncope. ^*^ Significant difference (*p* < 0.001) compared to the control group, as calculated using the Student’s *t*-test and the Chi-squared test.

**Table 2 ijerph-17-01908-t002:** Variations in repolarization parameters in children with vasovagal syncope, either with morphological changes on the electrocardiogram or without, compared to those with psychogenic pseudosyncope (control).

QT Variables		VVS I ^1^ (*n* = 19)	VVS II ^2^ (*n* = 11)	Control ^3^ (*n* = 30)	*p*	*p*	*p*
		Mean (SD), ms	Mean (SD), ms	Mean (SD), ms	(VVS I vs. VVS II)	(VVS I vs. Control)	(VVS II vs. Control)
	RR	843.2 (136.6)	864.6 (131.6)	860.7 (208.7)	0.553	0.748	0.955
**ECG0**	QT	380.0 (22.9)	375.5 (16.9)	370.0 (20.7)	0.582	0.120	0.439
	QTc	416.0 (14.8)	406.7 (29.6)	403.4 (29.3)	0.350	0.09	0.752
	QTp	289.5 (21.5)	285.5 (16.9)	290.7 (20.8)	0.445	0.848	0.462
	QTpc	316.6 (10.9)	309.1 (23.0)	316.9 (26.7)	0.326	0.956	0.395
	TpTe	90.5 (5.2)	90.0 (0.0)	79.3 (5.2)	0.832	<0.001 ^*^	<0.001 ^*^
	RR	827.4 (145.7)	790.9 (85.4)	861.3 (212.4)	0.457	0.544	0.295
**ECG1**	QT	385.8 (21.2)	377.3 (14.9)	384.7 (31.5)	0.251	0.892	0.461
	QTc	427.4 (25.5)	425.6 (22.4)	419.0 (27.0)	0.849	0.285	0.474
	QTp	294.7 (21.2)	288.2 (15.4)	306.0 (29.2)	0.378	0.153	0.062
	QTpc	326.2 (20.2)	325.0 (18.5)	333.1 (23.7)	0.870	0.301	0.313
	TpTe	91.1 (4.6)	89.1 (3.0)	78.7 (5.7)	0.420	<0.001 ^*^	<0.001 ^*^
	RR	595.3 (72.5)	650.9 (100.1)	635.3 (206.5)	0.089	0.421	0.813
**ECG2**	QT	347.4 (19.4)	349.1 (22.6)	348.0 (34.8)	0.827	0.943	0.924
	QTc	451.3 (13.4)	434.4 (15.4)	442.3 (20.3)	0.004^*^	0.092	0.253
	QTp	247.4 (19.1)	260.9 (22.1)	283.0 (31.4)	0.085	<0.001 ^*^	0.039 ^*^
	QTpc	321.0 (11.3)	324.3 (12.2)	359.4 (19.8)	0.470	<0.001 ^*^	<0.001 ^*^
	TpTe	100.0 (3.3)	88.2 (4.0)	65.0 (6.8)	<0.001^*^	<0.001 ^*^	<0.001 ^*^
	RR	743.7 (170.5)	787.3 (185.9)	788.3 (191.9)	0.519	0.412	0.987
**ECG3**	QT	368.4 (29.9)	365.5 (21.6)	368.0 (29.8)	0.776	0.962	0.797
	QTc	431.8 (25.5)	417.4 (31.1)	419.6 (28.2)	0.180	0.132	0.831
	QTp	275.8 (28.5)	275.5 (21.6)	290.3 (27.4)	0.973	0.081	0.113
	QTpc	322.5 (16.8)	314.1 (21.1)	330.8 (24.0)	0.239	0.162	0.049 ^*^
	TpTe	92.6 (4.5)	90.0 (0.0)	77.7 (5.7)	0.250	<0.001 ^*^	<0.001 ^*^

^1^ VVS I: children with vasovagal syncope (VVS) and a change in their T-wave morphology during the head-up tilt table test. ^2^ VVS II: children with VVS and no change in their T-wave morphology during the head-up tilt table test. ^3^ Control: children with psychogenic pseudosyncope. ^*^ Significant difference (*p* < 0.05) as calculated using the ANOVA, Bonferroni post-hoc test, and *U* Mann-Whitney and Student’s *t*-test.

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
