# Peer review of "The Repolarization Period during the Head-Up Tilt Test in Children with Vasovagal Syncope"

_ijerph, 2020, doi:10.3390/ijerph17061908_

Round 1

Reviewer 1 Report

In this retrospective study, the authors aimed to investigate whether ventricular repolarization parameters, measured during HUTT, could be indicative of future ventricular arrhythmias in children with syncope. They found that in children with VVS and HUTT negative, the ECG performed in the post tilt phase showed a change in the morphology of the T waves compared to the controls and QTc interval and Tpeak-Tend longer than the controls. They therefore speculated that children with these ECG characteristics have a hidden predisposition to develop arrhythmias "in the future".

Repolarization disorders have been previously described in both adult patients with VVS during HUTT and in patients already diagnosed with long QT syndrome. However, these were all retrospective studies and no one demonstrated the association of these changes with the subsequent appearance of ventricular arrhythmias or other adverse events. So, although the hypothesis suggested by the authors is intriguing, at the present state of the art, it remains purely speculative and of little usefulness in clinical practice.

Indeed, the authors are aware of this important limitation, as stated in the discussion.

Some points should be addressed:
1) Abstract: line 19- please specify overall number of cases (19/30 children dysplayed ...).
2) Methods: the authors do not specify whether ECG measurements have been carried out by one or more researchers and refer generically to a check by specialists. Please specify whether a double blind measurement has been made for all ECGs and, if not, why.
3) Results: Figures 1 and 2 exemplify the ECG abnormalities of the patients of the VVS1 and VVS2 groups. They present the ECG of a single representative patient for each group. Please specify better in the Title/legend figure.
4) Discussion-Conclusions. Since, as correctly stated by the authors (page 9 lines 325-328), the paper does not demonstrate the occurrence of arrhythmias but only of ECG abnormalities that could potentially predispose to the arrhythmias, the conclusions should be mitigated.

Finally, some typing errors should be corrected (for example some quotes scattered throughout the text - page 1, line 29 etc.)

Author Response

Dear Reviewer,

thank you for the very helpful and valuable suggestions. Below we explain point-by-point the corrected details of the revisions in the manuscript:

1) Abstract: line 19- please specify overall number of cases (19/30 children dysplayed ...).

- we agree and correct it as suggested

2) Methods: the authors do not specify whether ECG measurements have been carried out by one or more researchers and refer generically to a check by specialists. Please specify whether a double blind measurement has been made for all ECGs and, if not, why.

  • the ECG measurements were taken independently by 2 researchers and checked by a specialist in cardiac pediatrics. Measurement of repolarization parameters requires a lot of patience and precision, which is associated with the long duration of these measurements. The research project had to fit in the designated time frame where it was necessary to report on the stages of research advancement. We have lost the time associated with changing the direction of research. The initial assumption of the project was different, the group of children with the correct tilt test was to be a control group. When we saw the abnormalities in T-wave we could not leave this phenomenon indifferently and decided to change the concept of the project. That is the reasons why there was no double blind. It was difficult to gain an independent separate research team.

3) Results: Figures 1 and 2 exemplify the ECG abnormalities of the patients of the VVS1 and VVS2 groups. They present the ECG of a single representative patient for each group. Please specify better in the Title/legend figure.

  • thank you for your important attention, we will improve the work as you suggested.

4) Discussion-Conclusions. Since, as correctly stated by the authors (page 9 lines 325-328), the paper does not demonstrate the occurrence of arrhythmias but only of ECG abnormalities that could potentially predispose to the arrhythmias, the conclusions should be mitigated.

  • we agree with the review and we changed the conclusions and we made it more mitigated. The conclusion after changing are following: “we need further follow-up studies to establish the clinical importance of abnormal dynamics of the repolarization period in children with vasovagal syncope and negative HUTT results. From the clinical standpoint, the children with VVS and negative HUTT results, who are diagnosed with after-tilt repolarization abnormalities in ECGs, require systematic and frequent check-ups in the cardiology out-patient clinic in addition to standard prophylactic recommendations with lifestyle modifications”.
  1. Finally, some typing errors should be corrected (for example some quotes scattered throughout the text - page 1, line 29 etc.)

- we can’t see this scattered quotes and we don’t know what to improve.

Sincerely,

Ewelina Kolarczyk

Reviewer 2 Report

The authors demonstrated that the change in the T-wave morphology with prolongation of QTc and TpTe during HUTT may be potential arrhythmic risk especially ventricular tachyarrhythmias. The results of this study seem to be interesting and they proposed another aspect of HUTT for detecting latent long QT syndrome. However, there is some concern for acceptance.

Major points:

  1. As the authors described in limitation, none of the VVS I patients suffered from VT/VF or sudden death. It is very important limitation. This study needs to evaluate the clinical outcomes of these 19 patients in VVS I whether they have experienced ventricular arrhythmic events in the future. In addition, did you perform any other clinical evaluation for the risk of VT/VF such as Treadmill excise testing, pharmacological provocation using epinephrine or isoproterenol, or genetic analysis?
  2. In table 2, p-values comparing the difference of T-wave amplitude between lead V5 and II are necessary? Is there any meaning?
  3. Page 8, line 255-261: these sentences are not important. Please omit them.

Minor points:

  1. Page 1, line 20-21: QTc was significantly longer in VVS I patients compared to that in VVS II patients, but not in controls (p=0.092). Please correct this sentence.
  2. Page 1, line 24: because of no clinical evidence of arrhythmic events in this population, “might” seems to be suitable instead of “may”.
  3. In table 2, the range of T-wave amplitude of V5, ECG0 in VVS II shows 2.0-65. Is this value misprint?
  4. Page 8, line 232: the statistical value noted “p=0.04”, but table 3 shows the same value as “p=0.004”. Which is correct?
  5. The reference number 10 should move to before 23.

Author Response

Katowice, 9th March, 2020

Dear Reviewer,

thank you for the very helpful and valuable suggestions. Below we explain point-by-point the corrected details of the revisions in the manuscript:

Major points:

  1. As the authors described in limitation, none of the VVS I patients suffered from VT/VF or sudden death. It is very important limitation. This study needs to evaluate the clinical outcomes of these 19 patients in VVS I whether they have experienced ventricular arrhythmic events in the future. In addition, did you perform any other clinical evaluation for the risk of VT/VF such as Treadmill excise testing, pharmacological provocation using epinephrine or isoproterenol, or genetic analysis?

We have included the right suggestions in the conclusions that we need further follow-up studies to evaluate the clinical outcomes. The study group was strongly selected follow the including/excluding criteria which are detailed below. It was retrospective of patient’s medical records children which were hospitalized in children’s cardiology clinic. They have make the HUTT which Westminster Protocol without pharmacological provocation. 

  1. In table 2, p-values comparing the difference of T-wave amplitude between lead V5 and II are necessary? Is there any meaning?

In the study there were no statistically significant differences between the amplitude of the T wave in lead V5 and II. We wanted to show our work and comprehensive approach to this research. We delete the table according the suggestions of the review.

  1. Page 8, line 255-261: these sentences are not important. Please omit them.

According the review we have delete this sentences.

Minor points:

  1. Page 1, line 20-21: QTc was significantly longer in VVS I patients compared to that in VVS II patients, but not in controls (p=0.092). Please correct this sentence. – we changed it according the sugestion.
  2. Page 1, line 24: because of no clinical evidence of arrhythmic events in this population, “might” seems to be suitable instead of “may”. We agree, “might” – it’s better. We changed the conclusions: We need further follow-up studies to establish the clinical importance of abnormal dynamics of the repolarization period in children with VVS and negative HUTT. Therefore children with abnormal T-wave refraction and prolonged duration of the TpTe should remain under the care of a cardiological outpatient.
  3. In table 2, the range of T-wave amplitude of V5, ECG0 in VVS II shows 2.0-65. Is this value misprint? Yes, the dot is lost there. The correct is: 2.0-6.5
  4. Page 8, line 232: the statistical value noted “p=0.04”, but table 3 shows the same value as “p=0.004”. Which is correct? The correct is :0.004 (the one zero was missed)
  5. The reference number 10 should move to before 23. We fixed the reference: number 10 we moved to before 23

Sincerely,

Ewelina Kolarczyk
